# Structural Investigation of Orf Virus Bcl-2 Homolog ORFV125 Interactions with BH3-Motifs from BH3-Only Proteins Puma and Hrk

**DOI:** 10.3390/v13071374

**Published:** 2021-07-15

**Authors:** Chathura D. Suraweera, Mark G. Hinds, Marc Kvansakul

**Affiliations:** 1La Trobe Institute for Molecular Science, Department of Biochemistry and Genetics, La Trobe University, Melbourne, VIC 3086, Australia; C.Suraweera@latrobe.edu.au; 2Bio21 Molecular Science and Biotechnology Institute, The University of Melbourne, Parkville, VIC 3052, Australia

**Keywords:** poxvirus, orf virus, apoptosis, Bcl-2 proteins, X-ray crystallography

## Abstract

Numerous viruses have evolved sophisticated countermeasures to hijack the early programmed cell death of host cells in response to infection, including the use of proteins homologous in sequence or structure to Bcl-2. Orf virus, a member of the *parapoxviridae*, encodes for the Bcl-2 homolog ORFV125, a potent inhibitor of Bcl-2-mediated apoptosis in the host. ORFV125 acts by directly engaging host proapoptotic Bcl-2 proteins including Bak and Bax as well as the BH3-only proteins Hrk and Puma. Here, we determined the crystal structures of ORFV125 bound to the BH3 motif of proapoptotic proteins Puma and Hrk. The structures reveal that ORFV125 engages proapoptotic BH3 motif peptides using the canonical ligand binding groove. An Arg located in the structurally equivalent BH1 region of ORFV125 forms an ionic interaction with the conserved Asp in the BH3 motif in a manner that mimics the canonical ionic interaction seen in host Bcl-2:BH3 motif complexes. These findings provide a structural basis for Orf virus-mediated inhibition of host cell apoptosis and reveal the flexibility of virus encoded Bcl-2 proteins to mimic key interactions from endogenous host signalling pathways.

## 1. Introduction

Viruses utilize numerous immunomodulatory strategies to neutralize host cell apoptosis [1,2,3] to keep them viable for viral replication. Among the apoptosis regulators the B-cell lymphoma-2 (Bcl-2) family proteins are key to mitochondrial-mediated (or intrinsic) apoptosis [1,4]. The family of Bcl-2 proteins are recognized by the presence of up to four short sequence motifs, known as Bcl-2 homology (BH) motifs and are sub-divided into either pro-survival Bcl-2 or pro-apoptotic Bcl-2 proteins [5]. The pro-survival proteins in higher organisms include Bcl-2, Bcl-w, Bcl-x_L_, Mcl-1, A1 (Bfl-1) and Bcl-B [4] and are inactivated by the interactions with their pro-apoptotic Bcl-2 counterparts [5]. The pro-apoptotic Bcl-2 family are divided into two sub-groups: the multi-BH-motif pro-apoptotic Bcl-2 proteins, Bak, Bax and Bok; or the BH3-only proteins, which only feature the BH3 motif. The BH3-only group includes Bad, Bid, Bik, Bim, Bmf, Hrk, Noxa and Puma and apart from Bid are intrinsically disordered [5]. The BH3 region of the pro-apoptotic proteins interacts with the canonical ligand binding groove of pro-survival Bcl-2 proteins to neutralize their function [5,6]. This step is crucial to allowing pro-apoptotic Bak and Bax to launch apoptosis by permeabilizing the mitochondrial outer membrane (MOMP) [7,8] to release pro-apoptogenic factors such as cytochrome *c* from mitochondria that ultimately activate the caspase cascade to dismantle the cell [9].

Bcl-2 protein-mediated apoptosis appears to be evolutionarily well conserved [4] and Bcl-2 proteins are observed in early metazoans such as porifera (sponges) (*Geodia cydonium*) [10], placozoa (*Trichoplax adhaerens*) [11] and cnidaria (*Hydra vulgaris*) [12]. Along with metazoans, numerous large DNA viruses bear Bcl-2-like genes, reinforcing their pro-survival role by expressing cellular Bcl-2 mimics (viral Bcl-2 proteins, vBcl-2) [13]. These vBcl-2 proteins are sequence, structural, and functional homologs of apoptosis inhibiting mammalian Bcl-2 proteins [13]. Prominent virus-encoded Bcl-2 proteins include those from numerous poxviruses [2], but there are examples from many other viruses including Epstein Barr virus (EBV) BHRF1 [14,15], Kaposi-sarcoma virus (KSHV) KsBcl-2 [16], Adenovirus E1B19K [17] and African swine fever virus (ASFV) A179L [18]. Among these vBcl-2 genes some, for instance EBV BHRF1, adenovirus E1B19K, KSHV KsBcl2 and ASFV A179L, can be readily recognized by the presence of one or more BH motifs in their sequence [13]. On the other hand, most pox virus-encoded Bcl-2-like proteins lack detectable BH motifs despite the structural conservation of the Bcl-2 fold that comprises seven or eight α-helices, where α2–α5 combine to form the hydrophobic canonical ligand binding groove similar to that observed in cellular Bcl-2 proteins [2].

ORF virus (ORFV) is a large double stranded DNA virus belonging to the genus *para-poxviridae* [19], a sub-genus of the larger *cordopoxvirinae* sub-family, which consists of nine other sub-geneses that infect vertebrates [20]. ORFV is an emerging zoonotic disease that primarily infects sheep and goats and causes contagious ecthyma [21] that can be transmitted to humans by direct contact [22]. ORFV infections are characterized by proliferative skin lesions in the lips, nostrils, oral mucosa and muzzle of animals [21]. ORFV encodes ORFV125, a potent inhibitor of mitochondrial-mediated intrinsic apoptosis [23,24,25] and does not harbor recognizable Bcl-2 homology motifs [23,25]. However, ORFV125 bears a hydrophobic C-terminal region necessary for translocating the protein to mitochondria. ORFV125 supresses UV-induced DNA fragmentation, cytochrome *c* release and subsequent activation of caspases [23]. Structural and biochemical studies of ORFV125 revealed that it adopts a domain-swapped dimeric topology in solution [25] similar to that observed in some other poxviral Bcl-2 proteins such as Vaccinia virus (VACV) and variola virus (VARV) F1L [26,27,28], deerpox virus DPV022 [29] and tanapoxvirus TANV16L [30], and only interacts with four BH3 motifs of cellular pro-apoptotic Bcl-2 proteins: Bax, Bak, Puma and Harakiri (Hrk) [25]. Interestingly, ORFV125 did not show any detectable affinity towards Bim BH3 motif peptide, the universal human Bcl-2 interactor [25]. Here, we report the structural investigation of ORFV125 bound to Puma and Hrk BH3 motif peptides and propose that the mechanism of action of ORFV125 involves inhibition of these four pro-apoptotic proteins.

## 2. Materials and Methods

### 2.1. Protein Expression and Purification

Codon optimized synthetic cDNA encoding for ORFV125 (Uniprot accession number W6EVU4) but with the 30 C-terminal residues deleted was cloned into the bacterial expression vector pGEX-6P-1 (GenScript, Piscataway, NJ, USA). Recombinant ORFV125 was expressed in C41(DE3) cells using 2YT medium supplemented with 1 mg/mL ampicillin at 37 °C whilst shaking to an OD_600_ of 0.6. Isopropyl β-d-1-thiogalactopyranoside (IPTG) was added to a final concentration of 0.75 mM to induce protein expression for 18 h at 20 °C. Cells were harvested by centrifugation at 5000 rpm (JLA 9.1000 rotor, Avanti J E, Beckman Coulter, Brea, CA, USA) for 20 min and re-suspended in 100 mL of lysis buffer A (50 mM Tris pH 8.0, 300 mM NaCl, 5 mM DTT (dithiothreitol) and 1% Tergitol. The cells were lysed using sonication (programme 7, Model 705 Sonic Dismembrator, Fisher Scientific, Waltham, MA, USA) and lysates were centrifuged in SS34 tubes at 18,000 rpm (JA-25.50 rotor, Beckman Coulter Avanti J-E) for 30 min. The supernatant was applied to a 5 mL glutathione sepharose 4B column (GE Healthcare, Chicago, IL, USA) equilibrated with buffer A and washed with 150 mL of buffer A. HRV 3C protease was added overnight at 4 °C to liberate the target protein, which was subsequently eluted using buffer A and concentrated using a centrifugal concentrator (3 kDa molecular weight cut-off, Amicon^®^ Ultra 15) to a final volume of 3 mL. ORFV125 was subjected to size exclusion chromatography on a Superdex S200 10/300 column mounted on an ÄKTA Pure system (GE Healthcare) equilibrated in 25 mM HEPES pH 7.5, 150 mM NaCl and 5 mM TCEP (Tris(2-carboxyethyl)phosphine hydrochloride). The final sample purity was estimated to be greater than 95% based on SDS–PAGE analysis. Selected fractions were pooled and concentrated using a centrifugal concentrator with 3 kDa molecular weight cut-off (Amicon^®^ Ultra 15) to a final concentration of 6.4 mg/mL.

### 2.2. Crystallization and Structure Determination

Crystals for ORFV125: Puma BH3 and ORFV125: Hrk BH3 complexes were obtained by mixing ORFV125 with human Puma (Uniprot accession number Q96PG8, residues 130–155) or human Hrk 26-mer peptides (Uniprot accession number O00198, residues 26–51) that span the BH3 region of these proteins [31], using a 1:1.3 molar ratio as described [32] and concentrated using a centrifugal concentrator with a 3 kDa molecular weight cut-off (Amicon^®^ Ultra 0.5) to 6 mg/mL. Concentrated protein was immediately used for crystallization trials. Initial high throughput sparse matrix screening was performed using 96-well sitting drop trays (SWISSCI, Neuheim, Switzerland).

ORFV125: Hrk BH3 crystals were grown by the sitting drop vapour diffusion method at 20 °C in 0.2 M Sodium iodide, 0.1 M Bis-Tris propane pH 8.5, 20% PEG 3350. The crystals were flash cooled at −173 °C in mother liquor supplemented with 20% ethylene glycol. The ORFV125: Hrk BH3 complex formed single cuboidal crystals belonging to space group P1 with a = 46.40 Å, b = 57.69 Å, c = 65.06 Å, α = 71.88°, *β* = 76.60°, *γ* = 75.08° in the monoclinic crystal system.

All diffraction data were collected at the Australian Synchrotron MX2 beamline [33] using an Eiger 16M detector with an oscillation range of 0.1° per frame at a wavelength of 0.9537 Å. The diffraction data were integrated using DIALS [34] and scaled using AIMLESS [35]. A molecular replacement solution was obtained with PHASER [36] using the previously solved ORFV125: Bax structure [25] as the initial search model. The ORFV125: Hrk BH3 unit cell contained four molecules of ORFV125 and four Hrk BH3 peptides in the asymmetric unit, with a 40.3% solvent content and final translation function Z-score (TFZ) and log-likelihood gain (LLG) values of 46 and 6373, respectively. The final model of ORFV125: Hrk BH3 complex was built manually over several cycles using Coot [37]. Refinement was performed using PHENIX [38] with final R_work_/R_free_ values of 0.225/0.274, 99% of residues in the favoured region of the Ramachandran plot and no outliers. ORFV125: Puma BH3 crystals were obtained in 0.2 M magnesium acetate, 25% PEG 3350. The crystals were flash cooled at −173 °C in mother liquor supplemented with 20% ethylene glycol as cryoprotectant. The ORFV125: Puma BH3 complex formed single rod-shaped crystals belonging to space group P2_1_2_1_2_1_ with a = 48.87 Å, b = 69.44 Å, c = 91.28 Å, α = 90.00°, *β* = 90.00°, *γ* = 90.00° in the monoclinic crystal system. Diffraction data collection, integration and scaling were performed as described above. Molecular replacement was performed using PHASER [36] as described above. ORFV125: Puma BH3 crystals contain two molecules of ORFV125 and two Puma BH3 peptides in the unit cell, with a 38.79% solvent content and final TFZ and LLG values of 25.6 and 721.0 respectively. The final model of ORFV125: Puma BH3 was built and refined as above with final R_work_/R_free_ values of 24.5/28.5, and 99% of residues in Ramachandran favoured region of the plot and no outliers. All images for ORFV125: Hrk and ORFV125: Puma complexes were generated using PyMOL molecular graphic system version 1.8.6.0 (Schrödinger, LLC, New York, NY, USA). All raw images were deposited at the SBGridDB [39] using their PDB accession codes. All software were accessed through the SBGrid suite [40].

## 3. Results

We have previously shown that ORF virus-encoded Bcl-2 homolog ORFV125 has a restricted binding profile towards peptides spanning the BH3 motif (26-mer) of pro-apoptotic Bcl-2 proteins and binds only human Bak, Bax, Puma and Hrk BH3 motif peptides with nanomolar to sub micromolar affinities [25]. To investigate the molecular basis of the interaction with Puma and Hrk BH3 motifs with ORFV125 we crystalized ORFV25:Puma BH3 and ORFV125:Hrk BH3 complexes by reconstituting recombinantly expressed and purified ORFV125 with Puma and Hrk BH3 peptides. Crystals of ORFV125:Puma BH3 complex diffracted to 2.40 Å resolution, whereas ORFV125:Hrk BH3 complex diffracted to 2.00 Å (Table 1, Appendix A). Both complex structures were solved by molecular replacement using the previously determined structure of ORFV125:Bax BH3 (PDB ID 7ADT) [25] as a search model. For ORFV125:Puma BH3 complex a clear and continuous electron density map was observed for ORFV125 residues 5–61 and 65–133, and Puma BH3 motif residues 132–155 (Figure 1A,C), with the remainder of the residues presumed to be disordered. ORFV125 adopts the conserved Bcl-2 fold with seven α-helices in a domain-swapped dimeric configuration (Figure 2A) where the α1 helix of one protomer is swapped with the neighbouring protomer to form the dimer interface as previously seen in other poxvirus genera such as VACV F1L [27,28] (Figure 2C), VARV F1L [26], DPV022 [29] and TANV16L [30]. A structural similarity analysis was performed using DALI [41], which identified DPV022 (PDB ID 4UF1) [29] as the closest viral Bcl-2 homolog of ORFV125 with an rmsd of 2.6 Å over 119 Cα atoms and human Mcl-1 (PDB ID 3PK1) [42] as the closest cellular Bcl-2 homolog with an rmsd of 2.7 Å over 97 Cα atoms. Puma BH3 is bound to ORFV125 via the canonical hydrophobic ligand binding groove that is formed by helices α2–α5 (Figure 2A). The four canonical BH3 motif defining residues from Puma, I137, L141 and M144 and L148, engage four hydrophobic pockets of the canonical ORFV125 ligand binding groove (Figure 3A).

Furthermore, Puma Y152 occupies a fifth hydrophobic pocket in ORFV125, which is formed by residues A39, V42, W133 and V137 (Figure 4A). The hallmark of pro-survival Bcl-2 interactions with BH3 motifs is the presence of an ionic interaction between pro-survival Bcl-2 proteins and pro-apoptotic BH3 motif ligands between a conserved arginine in the BH1 motif and aspartate of the BH3-motif [4]. In ORFV125:Puma this interaction is formed between R87^ORFV125^ and D146^Puma^. An additional ionic interaction was observed between E80^ORFV125^ and R142^Puma^, and this interaction is supplemented by hydrogen bonds between E54^ORFV125^ and Q140^Puma^ as well as between G86^ORFV125^ and N149^Puma^ (Figure 3A).

The ORFV125:Hrk BH3 complex also showed clear and continuous electron density for ORFV125 residues 5–63 and 67–142 and Hrk BH3 motif residues 54–74 (Figure 1B,D). Similar to the ORFV125:Puma complex, ORFV125:Hrk BH3 adopted a domain-swapped topology (Figure 2B). The four conserved hydrophobic residues of the Hrk BH3 motif, T33, L37, L40 and L44, protrude into four hydrophobic pockets in the canonical, ORFV125 ligand binding groove. Furthermore, multiple ionic interactions were found between R87^ORFV125^ and D42^Hrk^, E80^ORFV125^ and R26^Hrk^, E54^ORFV125^ and R36^Hrk^, E46^ORFV125^ and R47^Hrk^. These salt bridges were supplemented by additional hydrogen bonds between R139^ORFV125^ amine group and the main chain carbonyl group of M49^Hrk^ (Figure 3B).

## 4. Discussion

Bcl-2 homologs are widely used amongst large DNA viruses to ensure viral proliferation and/or survival [2,13,43]. Amongst the *poxviridae,* the majority of genera have been shown to encode apoptosis inhibiting Bcl-2 homologs including the *orthopoxviridae* vaccinia virus F1L [27,28], variola F1L [26] and ectromelia viruses EMV025 [44], *leporipoxviridae* myxomavirus M11L [45], *cervidpoxviridae* deerpox virus DPV022 [29,46], *capripoxviridae* sheeppox virus SPPV14 [47,48], *avipoxviridae* fowlpoxvirus FPV039 [49] and canarypox virus CNP058 [50], *yatapoxviridae* tanapox virus TANV16L [30] and *parapoxviridae* orf virus ORFV125 [23,24,25]. Whilst many *poxviridae* encode for sequence, structural or functional pro-survival Bcl-2 homologs, considerable diversity exists amongst these proteins [2]. There are widely differing interaction profiles with host pro-apoptotic Bcl-2 and differences in overall structure and topology as well as detailed interactions at the atomic level, as might be expected from these highly sequence divergent Bcl-2-fold sequences [2,51]. Here we report the crystal structures of ORF virus-encoded Bcl-2 homolog ORFV125 in complex with BH3 motif peptides of host pro-apoptotic BH3-only proteins Puma and Hrk. A comparison of the interactions of Puma BH3 or Hrk BH3 with ORFV125 identified three conserved ORFV125 residues: E54, E80 and R87 in the binding groove which formed ionic interaction with the BH3 motif of pro-apoptotic Bcl-2 proteins. Among those interactions, the ionic interactions formed by ORF125 E80 and R87 are also conserved in the previously reported structure of ORFV125: Bax BH3 complex [25]. This suggests that ORFV125 residues E80 and R87 may be crucial mediators for BH3 motif peptide binding in the ORFV125 binding groove. Detailed analysis of the interaction between different poxvirus-encoded Bcl-2 proteins and host pro-apoptotic Bcl-2 reveals that whilst the BH1 motif is frequently not conserved in viral protein residues that occupy the structurally equivalent positions of the NWGR motif in the BH1 region of, for example, human Mcl-1 and Bcl-x_L_, it still performs important roles in the sequence divergent virus encoded Bcl-2 proteins. A comparison of ORFV125:Puma BH3 complex with TANV16L: Puma BH3 [30] reveals that the mode of interaction is nearly identical, despite the ORFV125:Puma interaction being approximately four-fold lower in affinity [25]. The hallmark interaction between the Arg residue from the NWGR motif and the Asp from the BH3 motif peptide is present in both complexes; the equivalent region of the BH1 motif in ORFV125 is formed by the sequence ‘SPGR’ whereas in TANV16L it is ‘NDNR’, with both contributing to binding and specificity [30] (Figure 3C). A similar observation was made for both sheeppox virus SPPV14 [48] and deerpox virus DPV022 [29], which utilize a different Arg to recapitulate the hallmark ionic interaction observed for mammalian pro-survival Bcl-2 proteins bound to BH3 motif bearing interactors [4]. These observations underscore the major importance of the BH1 equivalent region for determining interactions between Bcl-2 family proteins [4]. Mutational studies in the BH1 motif of human pro-survival Bcl-2 proteins abrogate BH3 binding [52]. Furthermore, ORFV125 and TANV16L complexes with Puma utilized the same four conserved hydrophobic residues of BH3 motif peptides to interact with four hydrophobic pockets of the binding groove. However, in ORFV:Puma BH3 a fifth hydrophobic pocket is used by residue Y152^PUMA^, a feature that has been previously observed for Bax complexes of Mcl-1, Bcl-x_L_ [42] and Bcl-2 [53], where Bax M74 is lodged in a comparable pocket. The engagement of a fifth hydrophobic pocket by Puma BH3 was unexpected, with comparable pocket engagement previously only seen for Bax BH3. A superimposition of the ORFV125 complexes of Bax BH3 and Puma BH3 shows that engagement is near identical (Figure 4A,B,D), whilst closely resembling the binding of the Bax BH3 M74 to Mcl-1 (Figure 4C). Beyond the use of a fifth hydrophobic pocket, Q140^PUMA^ forms a conserved hydrogen bonding interaction in both ORFV125 and TANV16L complexes. In contrast to Q140^PUMA^, R142^PUMA^ formed an additional ionic interaction with E80^ORFV125^ that is not seen in TANV16L:Puma BH3 complex (Figure 3A).

Puma is important for p53-dependent and independent regulation of apoptosis against various stimuli, such as genotoxic stress, radiation induced apoptosis, toxins, deregulated oncogene expression and viral infection [54]. Unlike other BH3-only proteins, Puma counteracts five cellular pro-survival Bcl-2 proteins (Bcl-2, Bcl-x_L_, Bcl-w, Mcl-1 and A1), though not Bcl-B [55] as well as most vBcl-2 proteins, through high affinity interactions with its BH3 motif [13]. A combination of structural data and knowledge of the conserved interaction between Bcl-2 proteins and Puma BH3 has been widely used for design peptidomimetic drugs to mimic these interactions [56]. In contrast to Puma, the molecular interactions supporting binding of BH3 motif peptide Hrk with ORFV125 are somewhat different to its previously reported interaction with sheeppox virus Bcl-2 homolog SPPV14 [48] (Figure 3D). The ORFV125:Hrk BH3 interaction is predominantly stabilized by four ionic interactions including the conserved ionic interaction formed by R87^ORF^ and D42^HRK^ (Figure 2B). In contrast, this ionic interaction found in the SPPV14:Hrk complex is not canonical and represents an alternative mode by which to form an ionic interaction utilizing D42 [48]. Interestingly, despite featuring an additional two ionic interactions compared to the Puma complex, the Hrk interaction with ORFV125 is of similar affinity (1912 nM for Hrk vs. 1753 nM for Puma) [25]; however, in the absence of a detailed mutagenesis analysis we cannot quantify the relative contributions to the affinity of the additional ORFV125:Hrk ionic interactions.

Despite adopting a domain-swapped dimeric configuration, the overall fold of ORFV125 closely resembles the conserved Bcl-2 fold and provides a hydrophobic groove for biding BH3-motifs. Complexes of ORFV125:Puma BH3 or ORFV125:Hrk BH3 are nearly identical to those observed for other Bcl-2:BH3 peptide complexes. A comparison of the structural similarity between ORFV125 with cellular counterparts using DALI analysis [41] showed that the closest mammalian structural homolog of ORFV125 is Mcl-1 (PDB ID 3PK1) [42] with an rmsd value of 2.7 Å over 97 Cα atoms, whilst sharing 19% sequence identity. However, several interesting differences are observed in the protein–peptide interfaces of ORFV125:Puma BH3 or Hrk BH3 complexes as well as Mcl-1:Puma BH3 complex and sheeppox virus SPPV14:Hrk BH3 complex. The sequence–structure alignment of ORFV125 with Mcl-1 revealed that the regions harbouring the BH motifs feature significant sequence variation, mostly centred near the binding groove, and provide the structural basis for the selectivity variation observed for pro-apoptotic BH3 ligands, with the ORFV125 binding profile towards BH3 ligands being more restricted and only binding to Bak, Bax, Puma and Hrk with sub-micromolar affinity [25]. In contrast, Mcl-1 showed affinity for a broad range of BH3 ligands and binds to Bak, Bax, Puma and Hrk tightly with low nanomolar affinity [57]. These data imply that ORFV125 inhibits apoptosis mediated by these four pro-apoptotic proteins. However, the relative contributions to the overall inhibition of premature host cell death made by each of the interactors remains to be clarified. Bax and Bak are vital to mitochondrial rupture and release of pro-apoptotic factors from this organelle, while the BH3-only proteins Puma and Hrk have more specific roles in initiating apoptosis [57]. Inhibition of the action of Puma and Hrk would allow the major cellular pro-survival proteins to maintain cell viability during invasion by ORF virus, with ORFV125 likely to work in tandem with endogenous pro-survival Bcl-2 proteins to maintain host cell viability. Our structures provide detailed mechanistic insights to provide a platform for deciphering the molecular interactions underlying inhibition of premature host cell apoptosis during ORF virus infectivity and proliferation.

## Figures and Tables

**Figure 1 viruses-13-01374-f001:**
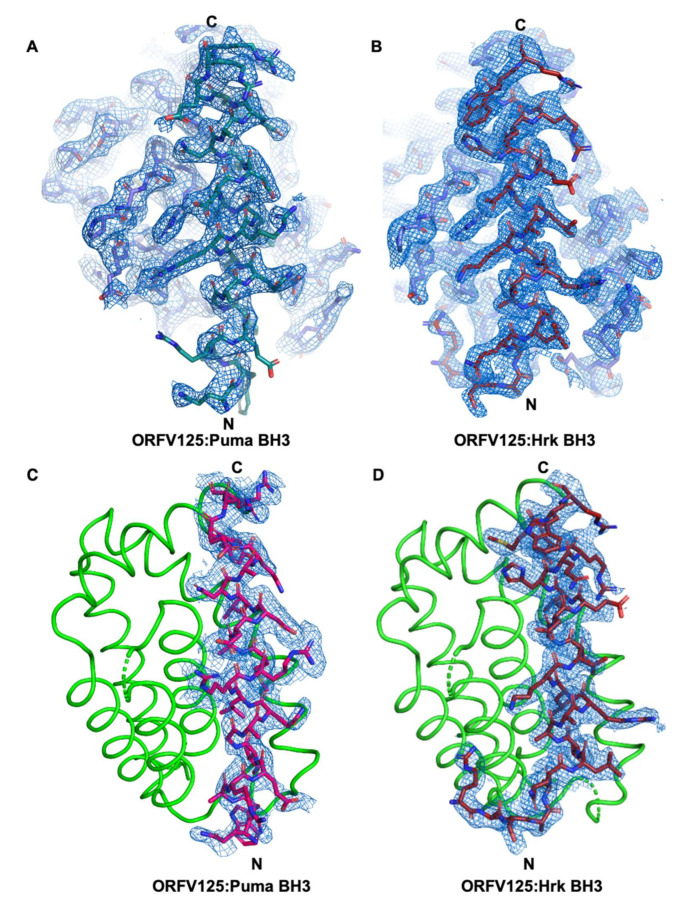
Electron density maps of ORFV125: Hrk and Puma BH3 complexes. 2Fo-Fc electron density maps of (**A**) ORFV125: Puma BH3 complex and (**B**) ORFV125: Hrk BH3 complex, interfaces contoured at 1.5 σ. The N- and C-termini of the BH3 peptide are indicated. (**C**) Omit map of ORFV125: Puma BH3 complex and (**D**) ORFV125: Hrk BH3 complex, contoured at 3 σ. BH3 peptides are shown as sticks, ORFV125 backbone is shown as cartoon.

**Figure 2 viruses-13-01374-f002:**
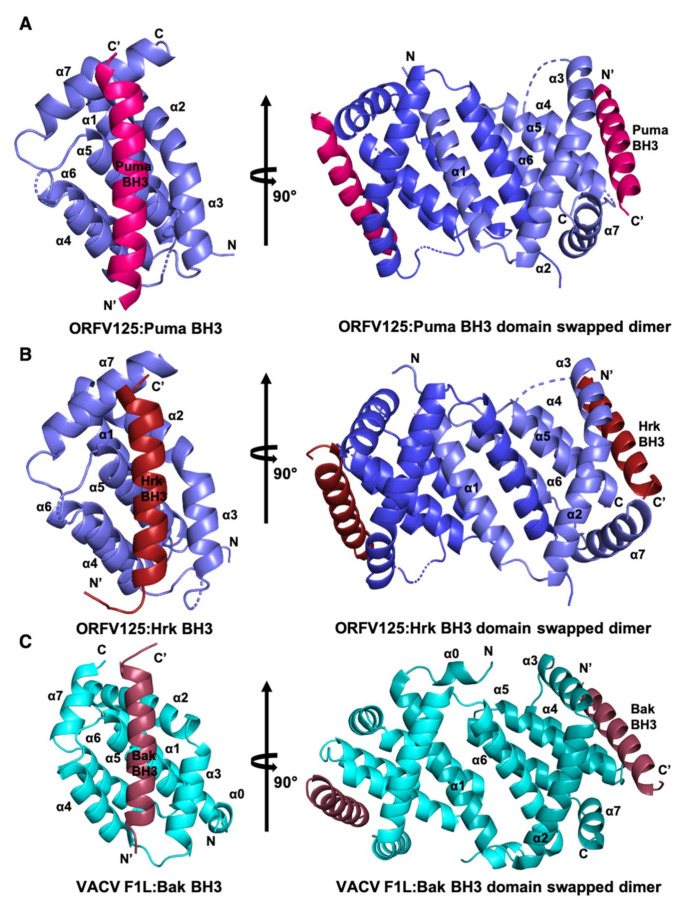
ORFV125 binds to BH3 motif peptides of proapoptotic Bcl-2 proteins using the canonical ligand binding groove. Crystal structures of ORFV125 bound to Puma and Hrk BH3 motifs. (**A**) ORFV125 (slate) in complex with the Puma BH3 motif (hot pink). ORFV125 helices are labelled α1–α7. The left-hand view is of the hydrophobic binding groove of one protomer formed by helices α3–α5, and the right-hand view is the domain-swapped dimer viewed along the two-fold symmetry axis between the domain-swapped α1 helices. (**B**) ORFV125 (slate) in complex with the Hrk BH3 domain (brick red) (**C**) VACV F1L (cyan) in complex with the Bak BH3 domain (raspberry). Images were generated using the PYMOL Molecular Graphics System, Version 1.8 Schrödinger, LLC. 6. The N- and C-termini are indicated as N and C for the Bcl-2 protein and N’ and C’ for the BH3 peptide, respectively.

**Figure 3 viruses-13-01374-f003:**
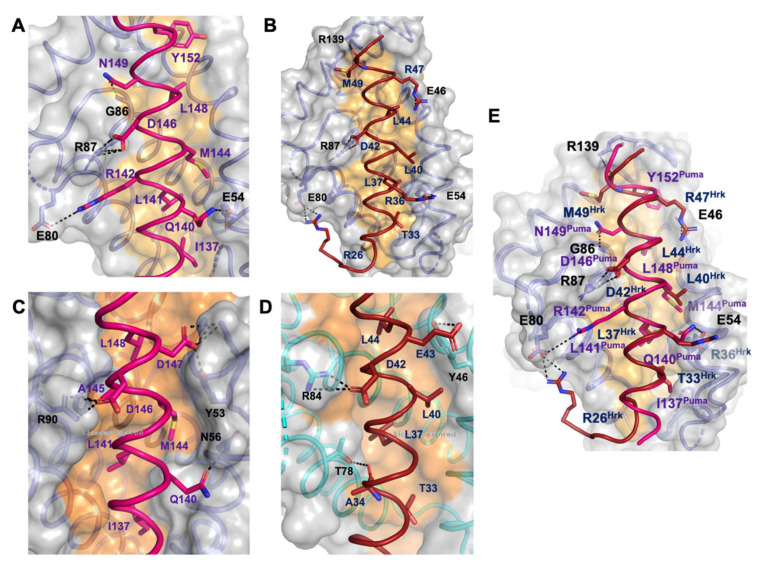
Detailed view of the ORFV125:Puma BH3, ORFV125:Hrk BH3, TANV16L:Puma BH3 and SPPV14:Hrk BH3 interfaces. The ORFV125 surface, backbone and floor of the binding groove are shown in grey, slate and orange, respectively. (**A**) ORFV125:Puma BH3 interface where Puma BH3 is shown in hot pink. The five key hydrophobic residues of Puma (I137, L141, M144, L148 and Y152) protruding into the binding groove, and the conserved salt bridge formed by ORFV125 R87 and Puma BH3 D146 are labelled, as well as all other residues involved in additional ionic interactions and hydrogen bonds. Interactions are denoted by dashed black lines. (**B**) ORFV125:Hrk BH3 with the surface of ORFV125 is shown as in (**A**), and Hrk BH3 is shown in brick red. The four key residues of Hrk BH3 (T33, L37, L40 and L44) are protruding into the binding groove, and the conserved salt bridge formed by Hrk D42 and ORFV125 R87 is labelled, as well as residues involved in hydrogen bonds. (**C**) TANV16 (sky blue):Puma BH3 is shown as in (**A**). The four key hydrophobic residues of Puma BH3 (I137, L141, M144 and L148) are protruding into the binding groove, and the conserved salt bridge formed by Puma D146 and TANV16L R90 is labelled, as well as residues involved in hydrogen bonds. Interactions are denoted by black dotted lines. (**D**) SPPV14 (cyan):Hrk BH3 is shown as in (**B**). The four key residues of Hrk BH3 (T33, L37, L40 and L44) are protruding into the binding groove, and the conserved salt bridge formed by Hrk D42 and SPPV14 R84 is labelled, as well as residues involved in hydrogen bonds. Interactions are denoted by black dotted lines. (**E**) Superimposition of Puma and Hrk BH3 complexes of ORFV125 as shown in panels (**A**,**B**). Key interacting residues are shown as sticks and are labelled.

**Figure 4 viruses-13-01374-f004:**
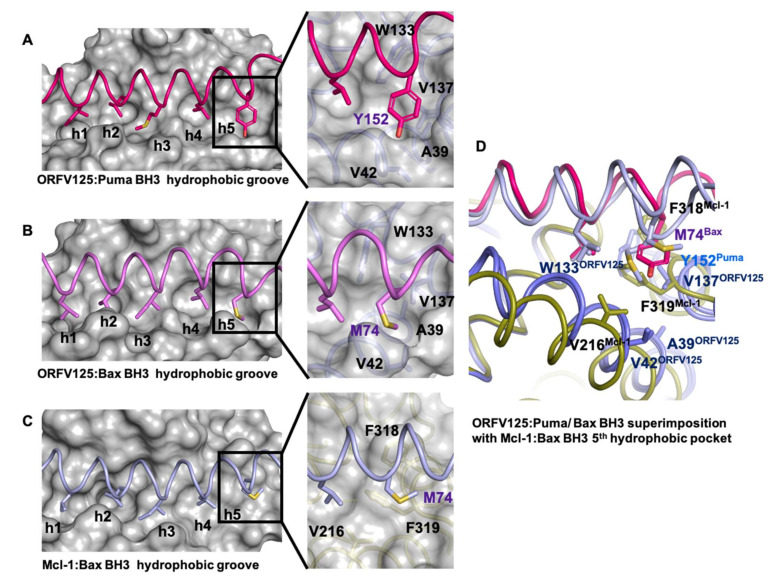
Engagement of the fifth hydrophobic pocket in ORFV125 and Mcl-1. (**A**) Hydrophobic binding groove of ORFV125:Puma BH3 complex. Five hydrophobic pockets in the binding groove are labelled h1-h5. The surface of the ORFV125 is shown in grey with side chains of the key residues in pocket h5 shown as sticks. Puma BH3 is shown as sticks (hot pink). An enlarged view of the fifth hydrophobic pocket of ORFV125 is shown in the middle panel with a cartoon tube of the ORFV125 backbone (slate), with residues involved in forming the fifth hydrophobic pocket labelled. (**B**) Hydrophobic binding groove of ORFV125:Bax BH3 complex (PDB ID 7ADT). ORFV125 is shown as in (**A**), Bax BH3 is shown as sticks (magenta). (**C**) Hydrophobic binding groove of Mcl-1:Bax BH3 complex (PDB ID 3PK1). The surface of Mcl-1 is shown in grey with side chains of the key residues in pocket h5 shown as sticks. Bax BH3 is shown as sticks (sky blue). (**D**) The enlarged view of a superimposition of the fifth hydrophobic pocket formed by ORFV (slate):Puma (hot pink) BH3, ORFV125 (slate):Bax (light blue) BH3 and Mcl-1 (olive):Bax (light blue) BH3. ORFV125 and Mcl-1 peptide backbones are shown as cartoon tubes, sidechains of the residues that form the fifth hydrophobic pocket are shown as sticks and labelled. Puma BH3 (hot pink) and Bax BH3 (sky blue) are shown as cartoon and sticks.

**Table 1 viruses-13-01374-t001:** X-ray crystallographic data collection and refinement statistics. Values in parentheses are for the highest resolution shell.

Data Collection	ORFV125: Hrk BH3	ORFV125: Puma BH3
Space group	P1	P2_1_2_1_2
Cell dimensions		
a, b, c (Å)	46.40, 57.69, 65.06	48.87, 69.44, 91.28
α, β, γ (°)	71.88, 76.60, 75.08	90.00, 90.00, 90.00
Wavelength (Å)	0.9537	0.9537
Resolution (Å)	47.03–1.99 (2.05–1.99)	49.03–2.50 (2.60–2.50)
R_sym_ or R_merge_	0.054 (1.46)	0.096 (1.35)
R_pim_	0.045 (1.09)	0.054 (0.74)
I/σI	6.4 (0.40)	7.8 (0.90)
Completeness (%)	96.7 (94.1)	98.1 (98.4)
CC_1/2_	0.99 (0.25)	0.99 (0.31)
Redundancy	2.7 (2.5)	4.1 (4.1)
Refinement		
Resolution (Å)	47.03–1.99 (2.05–1.99)	49.03–2.50 (2.60–2.50)
No. reflections	40,315	11,082
R_work_/R_free_	0.225/0.274	0.245/0.285
Clashscore	1.35	1
No. atoms		
Protein	4761	2479
Ligand/ion	38	0
Water	70	25
B-factors		
Protein	69.08	82.98
Ligand/ion	83.77	0
Water	67.07	74.2
R.m.s. deviations		
Bond lengths (Å)	0.006	0.003
Bond angles (°)	0.71	0.48

## Data Availability

Data supporting the findings of this manuscript are available from the corresponding authors upon reasonable request. Coordinate files were deposited at the Protein Data Bank (https://www.rcsb.org/) (accessed 1 July 2021) using accession codes 7P0U and 7P0S for ORFV125:Hrk BH3 and ORFV125:Puma BH3, respectively. The raw X-ray diffraction data were deposited at the SBGrid Data Bank [39] (https://data.sbgrid.org/data/) using their PDB accession codes 7P0U and 7P0S for ORFV125:Hrk BH3 and ORFV125:Puma BH3, respectively.

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
