# Peer review of "Structural Investigation of Orf Virus Bcl-2 Homolog ORFV125 Interactions with BH3-Motifs from BH3-Only Proteins Puma and Hrk"

_viruses, 2021, doi:10.3390/v13071374_

Round 1

Reviewer 1 Report

The authors have presented a structural basis for the interaction between ORFV125, an important Bcl2 protein, and interacting proteins Puma3 and Hrk4. The study is a useful addition to the literature, and will be of interest to the field. While the overall content and presentation is good, there a number of things that need be addressed prior to publication.

  1. A very thorough review for spelling and punctuation. A short list is presented here:
    1. Line 39: Missing full stop: [5,6] This step is
    2. Line 42: Rework "demolish the cell"
    3. Line 45: Missing full stop: [12]) Along with
    4. Line 50: Insert space: A179L[18].
    5. Line 58: Rework: ORFV is emergent zoonotic disease
    6. Line 64: Rework: it exists as domain swapped dimer in solution
    7. Line 66: Add space: [29]and tanapoxvirus
    8. Line 121: Remove space: For ORFV125: Puma BH3
    9. Line 139: Add space(PDB ID 4UF1)[29] as
    10. Line 140: Add space (PDB ID 3PK1)[42]
    11. Line 145: Rework: The hallmark ionic interaction between pro-survival Bcl-2 proteins and pro-apoptotic BH3 motif ligands between a conserved arginine in the BH1 motif and aspartate of the BH3-motif [4] is formed between R87ORFV125 and D146Puma. 
  2. The authors need to redo the figures where the word "licence expired" is visible
  3. The I/sig is very low. While I don't want to be pedantic over how to judge resolution cut-off as there are acceptable ways, across the board, CC1/2, I/sig, and R(merge) all suggest the resolution range that has been chosen MAY not that meaningful. Rather than suggesting to remove data, can the authors please state why they have chosen to extend the resolution to this range? Do the maps appear better for example?
  4. It would be somewhat useful to include Rpim as there are well reported issues with Rmerge. I'm not suggesting to remove Rmerge, this is also useful to report, but having Rpim is also useful, especially in the outer resolution shell.
  5. PDB validation reports were not provided. Most journals require these, and they are useful for the reviewer to examine.

Author Response

  1. A very thorough review for spelling and punctuation. A short list is presented here:
    1. Line 39: Missing full stop: [5,6] This step is
    2. Line 42: Rework "demolish the cell"
    3. Line 45: Missing full stop: [12]) Along with
    4. Line 50: Insert space: A179L[18].
    5. Line 58: Rework: ORFV is emergent zoonotic disease
    6. Line 64: Rework: it exists as domain swapped dimer in solution
    7. Line 66: Add space: [29]and tanapoxvirus
    8. Line 121: Remove space: For ORFV125: Puma BH3
    9. Line 139: Add space(PDB ID 4UF1)[29] as
    10. Line 140: Add space (PDB ID 3PK1)[42]
    11. Line 145: Rework: The hallmark ionic interaction between pro-survival Bcl-2 proteins and pro-apoptotic BH3 motif ligands between a conserved arginine in the BH1 motif and aspartate of the BH3-motif [4] is formed between R87ORFV125 and D146Puma. 

Response: We have addressed the spelling and punctuation issues listed, as well as several others that we identified during the revision process.

  1. The authors need to redo the figures where the word "licence expired" is visible

Response: We generated the figures without the watermark

  1. The I/sig is very low. While I don't want to be pedantic over how to judge resolution cut-off as there are acceptable ways, across the board, CC1/2, I/sig, and R(merge) all suggest the resolution range that has been chosen MAY not that meaningful. Rather than suggesting to remove data, can the authors please state why they have chosen to extend the resolution to this range? Do the maps appear better for example?

Response: The issue of resolution cut-off is always somewhat of a challenge. We opted in the end to go with CC1/2 at 0.25 and above based on the Karplus and Diederich (2012). We did not systematically cut the resolution at regular intervals and assessed fully refined maps, however the maps appeared visually better at CC1/2 0.25 than compared to traditional cut-offs such as R-merge 0.5 in the outer shell.

  1. It would be somewhat useful to include Rpim as there are well reported issues with Rmerge. I'm not suggesting to remove Rmerge, this is also useful to report, but having Rpim is also useful, especially in the outer resolution shell.

Response: We added the Rpim value in the crystallographic table

  1. PDB validation reports were not provided. Most journals require these, and they are useful for the reviewer to examine.

Response: We apologize for their absence. We did have the reports, but they were not requested at the submission stage.

Reviewer 2 Report

In this manuscript, Suraweera et al. describe the crystal structures of ORFV125 protein in complex with peptides harbouring the sequence for BH3 motif of Puma and Hrk, two proapoptotic Bcl-2 proteins. The manuscript builds on previous data published by the same laboratory, describing ORFV125 strcuture and binding affinities with Bax, Bak, Puma and Hrk.

The authors present a clear and convincing structural analysis to explain ORF virus inhibition of Bcl-2 mediated apoptosis in the host. Given the low sequence conservation among Bcl-2 proteins from various poxvirus, this study will make a nice contribution to the field.

Specific considerations:

- Protein expression and purification section (in Materials and Methods) should contain at least a short description of the procedure. I suggest mentioning cloned sequences, expression system, presence/removal of purification tags, purification strategy, ... This would make it easier for the reader to get an idea of the source of the sample, without having to dig in other papers.

- PDB codes for both structures (line 237) and Rwork/Rfree values for ORFV125: Puma BH3 structure (line 102) are missing. Please bear in mind that reviewers don't have access to the crystal structures presented in the manuscript, so having them deposited at PDB is the only way to ensure that at least they have gone through a validation process and that they meet the field standards.

- Given that both structures were solved using a search model which already included a BH3 molecule, consider adding an omit map for each structure (to Fig 1 or as SI), to ensure that the obtained density is not biased by the atomic model used for molecular replacement.

- Does crystal packing involve any contacts of Hrk/Puma BH3 molecules with other BH3 or ORFV125 molecules from neighbouring complexes? If so, it is important to state such contacts in the manuscript, since this could influence the conformation adopted by the BH3 peptides in the crystal structures.

- The source of the Puma & Hrk BH3 sequences should be better described in the Materials and Methods: are they from human or from sheep? Or do these domains share the same sequence in both species?

Other minor points:

- Some minor typos in the text need to be addressed:

  • lines 44-45: revise position of parenthesis in "early metazoans (such as ...)" and add a full stop at the end of the sentence.
  • line 86: Eiger 16M detector
  • line 94: consider splitting the paragraph to separe ORFV125:Hrk model building and ORFV125:Puma crystallisation.
  • line 97: "belong" should read "belonging"?
  • line 142: alpha symbol on helices a2-a5
  • line 211: review position of article "a" in "Q140PUMA forms a conserved a hydrogen bonding interaction"
  • line 214: lower case for p53 (protein)
  • line 233: "the molecular interactions (...) are somewhat different"
  • lines 247-248: review sentence starting with "However, there are several..."

- Figure 3 could be improved by adding a superposition of the BH3 motifs from Puma and Hrk (from the ORFV125:BH3 complexes in Fig3A and 3B).

- In figure 4D, one of the two Bax molecules seems to be incomplete.

-Shouldn't line 213 cite Figure 3, instead of Figure 2?

- Figures 3C and 3D are not cited in the text; consider citing them (I would suggest to cite Figure 3C after line 195-198 and Figure 3D after line 236-237).

Author Response

- Protein expression and purification section (in Materials and Methods) should contain at least a short description of the procedure. I suggest mentioning cloned sequences, expression system, presence/removal of purification tags, purification strategy, ... This would make it easier for the reader to get an idea of the source of the sample, without having to dig in other papers.

Response: We have now included a full protein expression and purification section.

- PDB codes for both structures (line 237) and Rwork/Rfree values for ORFV125: Puma BH3 structure (line 102) are missing. Please bear in mind that reviewers don't have access to the crystal structures presented in the manuscript, so having them deposited at PDB is the only way to ensure that at least they have gone through a validation process and that they meet the field standards.

Response: We have now added the PDB IDs for both structures.

- Given that both structures were solved using a search model which already included a BH3 molecule, consider adding an omit map for each structure (to Fig 1 or as SI), to ensure that the obtained density is not biased by the atomic model used for molecular replacement.

Response: we have now added an omit map for each structure as suggested in a revised Figure 1.

- Does crystal packing involve any contacts of Hrk/Puma BH3 molecules with other BH3 or ORFV125 molecules from neighbouring complexes? If so, it is important to state such contacts in the manuscript, since this could influence the conformation adopted by the BH3 peptides in the crystal structures.

Response: Yes, this is the case. Puma BH3 residues R142, R143 and R154 are involved in crystal contacts with a neighbouring ORFV125 chain. However, the second copy of the Puma BH3 is not involved in crystal contacts, and no discernible differences are observed in the mode of engagement of one or the other copy Puma BH3 by ORFV125. For Hrk BH3, four copies are found in the structure, bound to four matched chains of ORFV125. Two copies of Hrk BH3 contact each other, however the other two copies are no involved in crystal contacts. Again no discernible differences are observed in the mode of engagement of the four Hrk BH3 chains by ORFV125. Since we had the benefit of multiple copies of the relevant BH3 peptides in each crystal structure, and in both cases were able to compare chains involved or free of involvement of crystal contacts, we did not elaborate on these contexts since they did not have a material effect on the structure of the BH3 peptide and the manner in which ORFV125 was engaged.

- The source of the Puma & Hrk BH3 sequences should be better described in the Materials and Methods: are they from human or from sheep? Or do these domains share the same sequence in both species?

Response: We now specify the species of the BH3 motif peptide sequences, which was human.

- Some minor typos in the text need to be addressed:

  • lines 44-45: revise position of parenthesis in "early metazoans (such as ...)" and add a full stop at the end of the sentence.
  • line 86: Eiger 16M detector
  • line 94: consider splitting the paragraph to separe ORFV125:Hrk model building and ORFV125:Puma crystallisation.
  • line 97: "belong" should read "belonging"?
  • line 142: alpha symbol on helices a2-a5
  • line 211: review position of article "a" in "Q140PUMA forms a conserved a hydrogen bonding interaction"
  • line 214: lower case for p53 (protein)
  • line 233: "the molecular interactions (...) are somewhat different"
  • lines 247-248: review sentence starting with "However, there are several..."

Response: We have addressed all typographical and grammatical issues specified.

- Figure 3 could be improved by adding a superposition of the BH3 motifs from Puma and Hrk (from the ORFV125:BH3 complexes in Fig3A and 3B).

Response: We have added a superposition as requested in an amended Figure 3 E

- In figure 4D, one of the two Bax molecules seems to be incomplete.

Response: This was an inadvertent problem with the chain selection, we now show the full trace for the second Bax chain in a revised Figure 4D.

-Shouldn't line 213 cite Figure 3, instead of Figure 2?

Response: Yes it should, we amended this.

- Figures 3C and 3D are not cited in the text; consider citing them (I would suggest to cite Figure 3C after line 195-198 and Figure 3D after line 236-237).

Response: We have called out the figures in the suggested locations.